# Beyond Colonial Politics of Identity: Being and Becoming Female Youth in Colonial Kenya

Elizabeth Ngutuku [1,2,*] 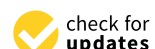 and Auma Okwany [3]

1. Firoz Lalji Institute for Africa, The London School of Economics and Political Science, Houghton Street, London WC2A 2AE, UK
2. Faculty of Humanities, University of Johannesburg, P.O. Box 524, Auckland Park 2006, South Africa
3. International Institute of Socal Studies of Erasmus University Rotterdam, Kortenaerkade 12, 2518 AX The Hague, The Netherlands; okwany@iss.nl
* Correspondence: e.m.ngutuku@lse.ac.uk

**Abstract:** This paper draws on biographical research among the Akamba and the Luo communities in Eastern and Western Kenya, respectively. Our research explored how practices of adolescence as a process, an institution, and a performance of identity interact with colonial modernities and imaginaries in complex ways. The biographical research was carried out predominantly with women born in the late colonial period in Kenya. We provide critical reflections on the process and affordances of our embodied storytelling approach, which we position as an Africanist methodology and a decolonial research practice. This research and approach provided women with a space to narrate and perform their lived experience, potentially disrupting epistemic inequities that are embedded in the way research on growing up in the past is carried out. The discussions show how colonialism interacted with other factors, including gender and generational power, tradition, girls' agency, and other life characteristics like poverty and family situation, in order to influence the lived experiences of women. Going beyond the narratives of victimhood that characterise coming of age in similar spaces, we present women's emergent, incomplete, and incongruent agency. We position this agency as the diverse ways in which people come to terms with their difficult contexts. The discussion also points to the need for unsettling the settled thinking about girlhood and coming of age in specific historical spaces in the global South.

**Keywords:** becoming; identity; colonial girlhood; storytelling; ubuntu; coming of age



## 1. Introduction: Re(Storying) Gendered Coming of Age in the Colonial Context

This paper draws on biographical research in Kenya in 2022 among the Akamba and the Luo communities in Eastern and Western Kenya, respectively. Carried out predominantly with women born between 1940 and 1963, this research explores how practices of adolescence as a stage, a process, an institution, and a performance of identity interact with colonial modernities and imaginaries in non-linear ways. We seek to accomplish two things in this paper. First, we present our critical reflections on this research process, which we see as an example of decolonial research practice. Second, drawing from women's accounts provides a perspective on the lived experience of being, becoming, and belonging as an adolescent girl during the period. Further, we argue that these processes of coming of age reflect women's sense of belonging and selfhood (who they were then and now), and how these are related to their identity and subjecthood (Willemse 2014).

As two long-term research collaborators and scholar-activists who have worked and lived in these communities, this research was borne out of our long-term commitment to decolonise knowledge and practices of knowing in relation to children and young people in scholarship, policy, and practice. We were also inspired by a desire to privilege epistemologies of the South (Okwany and Ngutuku 2023; Okwany and Ebrahim 2018; Okwany 2016; Santos 2014). In responding to the call for a Special Issue on the theme

decolonising East African genealogies of power, and as reflected in our decades-long collaborative work, we are inspired by the view and formulation by Mignolo (2020, p. 615) that decoloniality is 'an orientation to a praxis of living' and not just a critique. This understanding of decoloniality fits the role of our work and this research. For years, we have engaged the dominant ways of knowing about children and wellbeing, or coloniality of knowledge, through action-oriented research like what we carried out for this paper. We take a cue from Maldonado-Torres (2007, p. 243), who sees coloniality as 'patterns of power that resulted from colonialism and that have survived in the production of contemporary culture, knowledge, and the economy.' These forms of coloniality continue to influence how youthhood, both in colonial contexts and contemporaneously, is imagined, researched, and written about (see also Williams and Molebatsi 2019). This coloniality is also reflected in some oppressive research practices that are often extractive, revealing a need to decolonise our research methods, as we have attempted to do in our research. Linda Tuhiwai Smith comments on the coloniality of research by noting that research about indigenous people and, in our case, historical communities from the global South, draws from the collective memory of imperialism. Her argument that this memory determines how knowledge is collected and represented to those in the West, through their filters and back to the colonised, is useful for our approach (Tuhiwai-Smith 2013, pp. 1–2).

We are also influenced by Thambinathan and Kinsella (2021, pp. 1–2), who notes that 'decolonizing research means centring concerns and world views of non-Western individuals, and respectfully knowing and understanding theory and research from previously [Other(ed)] perspectives'. In most historical accounts of coming of age in Africa, colonialism is often represented as the central node that engendered specific experiences. While informative, such accounts eschew other nodes in the processes of coming of age. In reimagining this experience beyond the role of colonialism, the larger research from which we draw focused on the messy entanglements in the processes of coming of age. We explore how encounters between colonial and other power structures, like patriarchal and generational power, tradition, young people's agency, and other nodes, influenced this experience. In moving beyond the mere deconstruction of knowledge, our concern was not how young people's lived experience deviates from the existing normative and dominant accounts of coming of age and other colonial categories. Instead, by drawing from the perspectives of women who lived in this period, we sought to understand the diverse forces that animated young people's experience of what it meant to be a female youth and what this experience looked like in their everyday lives. Such accounts that frame young people as historical actors also aid in understanding the processes that erase such forms of agency. Our overall argument is that such erasures of the voices of women who lived in these contexts are a form of epistemic injustice that needs to be engaged with through empowering, transformational and reflexive research practices, like the storytelling approach used in our research. Our approach also decentres colonial narratives as the dominant factor in the women's lives and childhoods, gives agency to their stories, cultures and communities beyond these colonial/imperial structures.

We proceed with some key observations. While adolescence is often understood as a biological stage of transitioning to adulthood, our reference to adolescence is cautious, and we do not just see it as a stage. Some authors have noted that adolescence, like other social categories such as gender, is a social construct. For example, Petrone et al. (2015, p. 509) argue that 'how adolescence and youth are understood is always contingent on and constituted through social arrangements and systems of reasoning available within particular historical moments and contexts'. We therefore position adolescence as a contextualised and fluid process of belonging, which, in addition to being gendered, is also connected to people's wellbeing and relations with others, animals, places, and things.

Additionally, we do not intend to present these women's stories and accounts as a form of collective memory of their communities. Indeed, the two study communities can be found in several other counties in Kenya, and perceptions could differ based on location. Further, there is already some historical work that has been undertaken around these

communities that reflect the processes of growing up. However, authors like Musandu (2012) have argued that the first codifiers of Luo customs, such as historian Bethwel Ogot, wanted to preserve it for posterity and aimed at showing how the ancestors lived. Musandu (2012, p. 541) also contends that this work was bent on constructing a single Luo identity and assumed that customs were immutable before encounters with the West. Despite this proviso, we are also aware that the stories of these two communities, and communities in Africa in general, cannot be separated from the stories of the people who make up these communities, including these women.

Given the previous work done in this area (see, for example, Tamale 2011 and Nzegwu 2011, who have written on this topic from a decolonial perspective) and in these communities, we also do not purport to discover these women or authorise them. We also do not intend to talk about them in untold ways or do what van Wingerden sees (van Wingerden 2022, p. 3) as 'firsting', wherein a researcher claims first insights on something. Such claims to discovery would be erasing other histories, beings, and processes involved in intersubjective, inter-corporeal knowledge production. Instead, we set out primarily to affirm women's ways of knowing by giving women a platform to share their stories.

In telling these women's stories, our intention is not to reinvent or revise adolescence, or what Oruka saw as turning back the hands of time (cited in Masolo 2016), nor do we aim to reproduce the historical adolescent girl for our contemporary understanding of girlhood. Instead, an essential contribution of our work is the argument that memory can be political, and re-membering, a double word, that implies bringing to memory and bringing together the fragments of knowledge that have been excised through dominant ways of knowing and telling is an important component in decolonising knowledge (Mucina 2011a; Ngutuku and Okwany 2023; Ogot 1976). We are also inspired by the views of Williams and Molebatsi (2019, p. 113), that re(storying), an essential contribution from our research, can be a response to erasure and can show a presence or a protest.

Narration (like the stories of women we present) as an intervention into inequalities in knowledge production is also an end in and of itself, as well as a contribution to decolonising strategies. For example, Saidiya Hartman, an African American historian and cultural scholar, positions the act of narration as its own gift, an end in and of itself, and a form of reparation for what has been lost (Hartman 2008, p. 4). We also have a different kind of investment in telling the stories of these women. As authors [women] from Africa engaging with epistemic inequities in knowledge production, as we narrate the women's stories and reclaim their voice, we also reclaim our own voice and our relationships to these histories. We are inspired by the imperative of situated accounts embodied in the African proverb that asserts, 'Until lions have their historians, tales of the hunt shall always glorify the hunter' (Ngutuku and Okwany 2023, p. 256).

Our arguments are structured as follows: we provide perspectives on how girlhood and coming of age in colonial Africa are located within the dominant literature. We then contextualise notions of identity, being, and resilience as becoming and as they apply to processes of coming of age. This section is followed by our reflections on the process and affordances of our methodological approach, storytelling, which we position as an Africanist methodology and an embodied research approach. We then focus on local notions and practices of coming of age, drawing from select practices. Some of these include retrospective perspectives on girls' sexuality, and processes of guidance and responsibilisation. We follow this with a detailed exploration of the experience of coming of age through marriage. Marriage practices in these two communities have received considerable commentary, with the bulk of this literature drawing from what is seen as 'the historical legacy of racist fascination with alleged African profligate sexuality' (Mama 1996, as cited in Arnfred 2015, p. 150; Jacobs 1961). Before concluding, we offer a temporal temporal perspective on the identity of a female youth as becoming, as lived and remembered, by presenting perspectives on the futures imagined by these women, including reading their being, becoming and longing to belong from the present.

## 2. Being and Becoming Adolescents in Colonial Kenya: The Invisible, Docile, Female Youth

This section explores girlhood and coming of age in colonial Kenya. Several authors have commented on the hypervisibility of black male youth in colonial encounters that thrived on the exclusion of female youth, arguing that youth was quintessentially a male category in colonial thought (Ngutuku 2020). In the literature, these gendered binaries were seen as necessary in the project of colonial control and, subsequently, in the processes of decolonisation. For example, writing about the absence of girls in discourses of insurgency in colonial Kenya, Hynd (2021, p. 539) argues that most accounts written during this period were more about the youth, termed 'ado-combatants', primarily involved in insurgencies. Ngutuku (2020, p. 90) has also argued that within the discourse of black, male, youth as a threat to colonial authorities, the corresponding docile, female youth did not warrant much attention. Further, Ocobock (2017) argues that the elder state, or a state already 'maturing' and faced with uncertainties, used youth, specifically male youth, for control.

Girlhood is also entangled with womanhood in terms of relations of power. In cases where perspectives on the experience of women and girls are included, narratives of coming of age are often presented through the prism of victimhood to tradition and other perilous forms of being (see Ngugi 2009). Our arguments, therefore, respond to the need to place the views of those affected at the centre of our analysis and to view this experience in its own right, rather than putting it in relation to modernity, as has often been the case (Hurlbut 2015). For example, commenting on notions of girlhood and encounters with modernity as depicted in Ngugi wa Thiongo's novel, The River Between, Macharia (2012, p. 2) argues that encounters with colonialism and modernity among the Agikuyu of Kenya were seen as producing improperly gendered girls that threatened the ethnic intimacy, which had thrived on gender differentiations.

Among the Luo, women were often represented as victims of tradition in historical accounts. For example, Musandu (2012, p. 541) notes that the British norms that guided most of these writings could not comprehend female agency. Most of the literature on girlhood and the sexuality of women in this period, therefore, fails to consider the different ways in which vitality, agency, vulnerability, and hopeful futures also existed side by side in the processes of growing up. While we do not purport to erase some of the violence around the sexuality of women, we find company with Arnfred (2015), who argues that sexual violence was not the only way of being for women in these spaces. There is, therefore, a need for alternative narratives. We also provide a re-reading of this agentic girl by showing the contradictory and dynamic ways their agency manifested as part of their being and becoming as adolescents in these spaces. In our arguments, these forms of resistance and agency, or what we see as forms of resilience, do not always need to be read through oppositional realities or resistance to dominant power (Hughes 2022). Instead, we are interested in what is seen as vernacular forms of resilience by girls and women during this period, or resilience that is 'embedded in social practices and cultural repertoires' (Wandji et al. 2021, p. iii).

The experience of girlhood from the girls' perspective is also often missing from historical accounts. For example, Bellows-Blakely (2020, p. 7) argues that the experience of girlhood is often treated as a footnote in historical accounts, noting that 'historical explorations of girlhood in Africa rarely include the voices of people defined as girls.' Indeed, writing about female agency among the Luo, Musandu (2012) notes that much of the work on the history of girlhood has relied on those seen as knowledgeable, specifically elderly male gatekeepers. For example, the famous Sage/Sagacity Philosophy Project, pioneered by Odera Oruka of the University of Nairobi, was accomplished by interviewing people seen as folk sages or those who could recount Luo traditions (Masolo 2016). The only woman sage that Oruka interviewed, as noted by Mosima (2018, pp. 30–31), was seen as a 'timeless victim of a ferocious patriarchal order that Oruka does not explicitly interrogate'.

Similarly, even though the work of Evans-Pritchard, the renowned 20th-century anthropologist, is credited for going against the prejudicial accounts of his time, his work

among the Luo is still seen as conforming to the 'gendered, masculine, male gaze of most anthropologists' work' (Ngutuku and Okwany 2023, p. 256). We find this a fair assessment since, for example, his book on Luo Marriage customs (Evans-Pritchard 1950) was based on the testimony of one pastor, Ezekiel (Morton 2020, p. 199). By drawing on the life histories of several generations of women and, to a limited extent, re-enacting the traditional/colonial spaces and places of adolescence, our research heeds Nyamnjoh's call to bring such dominant accounts into dialogue with the people in question (Nyamnjoh 2012, p. 67).

In a similar vein, the processes of growing up in these communities have been the subject of much anthropological and quasi-anthropological work. For example, in his book, *Ethnology of Akamba and Other Tribes*, Hobley (1910), a colonial administrator in the Ukambani and Luo Nyanza regions, claimed the first systematic study of the Akamba people. Despite his disclaimer that he wanted to be loyal and accountable in his representation, his work reads more as an account of the exotic, the bestial, and the grotesque that characterises most of the Western gaze on the African way of life during this time (see Nkrumah 1963). For example, 'the gross' in the tenor of his work is seen in the negative way he represents some of the practices of coming of age, like body and teeth tattooing, associating some with cannibalism. Such disturbing anthropological accounts are often seen as products of their time. In response to such accounts, our work, aimed at enhancing a greater appreciation of the complexity of people's experiences and their histories, engages those practices that 'extract peoples from their contexts'. It is also geared towards humanising the stories of the Other. Women's accounts of coming of age should also be seen as corrective of the narratives of gender in the empire and historical accounts.

Most historical explorations of coming of age rely on archives (Hynd 2021; Vince et al. 2007; Kilonzo and Akallah 2021). However, Stoler (2002, p. 88) has argued that archives are imbued with unequal relations of power in knowledge production. She adds that archives, as representative of colonial legacies and as monuments of the state, were contingent on what could be written about in that space and time (Stoler 2002, p. 96). By drawing on the perspectives of women, our critique also creates, because it attempts to rework that which has been excluded from mattering, the perspectives of girls and women who lived in this context (Barad 2007, p. 235; Foucault 1988, p. 154; Spivak 1999; Tuhiwai-Smith 2013).

## 3. Coming of Age: Resilience Being, Belonging, and Becoming an Adolescent Girl

In this section, we expound on our understanding of youth, identity, becoming, and belonging, as well as agency and resilience, as they emerge from the lived experiences of the women in our research. By locating youthhood in lived contexts in our study, we perceive youth as an assemblage that draws connections between African relationalities of coming of age, generational relations, discourses and practices of youthhood, young people's agency, and how these connect to human and non-human others (Hynd 2021; Rozmarin 2021). Female youth is thus not a fixed category, its meanings shift over time and space, and age as a system of power relations complexly interacts with gender. Our analysis aligns with the argument by Loew (2012, p. 976) that age, in some contexts, can undercut gender, the socially constructed relations between men and women. She also asserts that 'we cannot understand gender without reference to its temporality.'

Belonging as part of young people's identities is also complex. Rozmarin (2021, p. 29) sees belonging as a 'complex and multifaceted relationship between people and their surroundings'. She notes that belonging is dimensional and includes belonging to a country, a place, a generation, a tradition, or a political group, among others. Belonging, therefore, has affective or emotional perspectives, and is associated with a longing to be at the right place, with the right people, or being at home (Antonsich 2010; Ngutuku 2023; Oostveen 2019). These forms of identification and subjecthood also materialise through interactions with non-human others. As we reveal, these non-human others include places, spaces, time, animals, beliefs, and gendered discourses (Barad 2007; Gabi 2013, p. 13; Rozmarin 2021; Taguchi 2011).

For a category like female youth, seen predominantly through the prism of vulnerability, processes of coming of age, belonging, and identity are entangled with their agency. This agency is what we present as their resilience or their sense of becoming (Irvinson and Renold 2021; Willemse 2014). Resilience involves strategies people use to come to terms with challenging contexts or becoming. Deleuze (1995, p. 170) sees becoming as 'individual and collective struggles that people undertake to come to terms with events and intolerable conditions and to shake loose, to whatever degree possible, from determinants and definitions'. Young people's resilience should also be understood through diverse temporalities, which are informed by a people's past, their views about the future, and how these are lived in the present. Rozmarin (2021, p. 364) supports this complex understanding of resilience and argues that becoming as complex gendered, cultural assemblages also 'emerge through ongoing practices entangled with place, history and landscape'. In this paper, some of the factors in these assemblages include the colonial project of control, gender, age, culture, tradition, rituals, and other such practices, and changes over time and space. As we demonstrate in our arguments, and drawing from Hughes (2022, p. 544), the subject of girlhood as becoming is therefore shaped by what is seen as 'intensities of life in all its forms.' These grounded understandings of resilience, identity, becoming, belonging, and being were also performed in contexts of interactions with the women through our research approach, which we now turn to.

**4. Methodological Approach: Storytelling and Performance as Africanist Methodologies**

> 'In Africa, everything is a story, everything is a repository of stories. Spiders, the wind, a leaf, a tree, the moon, silence, a glance, a mysterious old man, an owl at midnight, a sign, a white stone on a branch, a single yellow bird of omen, an inexplicable death, unprompted laughter, an egg by the river are all impregnated with stories.' (Okri 1997, p. 115)

We carried out this research among the Akamba and Luo communities in Eastern and Western Kenya, where we were born and have lived, as part of these communities' stories. We have also carried out research and work on childhood and youthhood in both communities since 2000. This research, whose ethical clearance was provided by the ethical committee of the London School of Economics and Political Science, was carried out between December 2021 and June 2022. We explored the shifting interstices around which the experience of being, becoming, and belonging as an adolescent girl was enacted in colonial Kenya. By using interstices, a spatial metaphor which implies a complex middle and encounters, this research was geared towards understanding the complex and emergent factors that interacted in multifaceted ways with young people's experiences of growing up in colonial spaces (Brighenti 2013). This research was not comparative, but was geared towards understanding how coming of age was experienced by the women in these communities and, where possible, offering perspectives on resonances between the two communities. Among the Akamba, this research was conducted in three villages in the central part of Kitui County, and among the Luo, in Siaya County, in three locations: Jera, Central Alego, and Gem.

Inspired by Mucina's work on Ubuntu relationalities that focused on his life (Mucina 2011b, p. 223), we believe that our personal stories are researchable. Our research was, therefore, inspired by the memoirs of our mothers, which we have been working on for several years. The storytelling by our mothers and the nature of this storytelling, their resilience and tenacity in the interstitial spaces of the colonial context, told to us as we grew up, fuelled our interest in the stories of adolescence in colonial Kenya. Therefore, we designed a larger study in order to incorporate more women's perspectives from the network of our mother's interpersonal relationships and wider contexts. This aimed to validate and affirm the issues that emerged from their biographical narratives.

Given our investment in reframing narratives of coming of age, our approach in this study was decolonial and aimed at centring women's views. We therefore used methods that enabled us to engage the inequities in knowledge production. Gallagher and Greenblatt

(2000, p. 50) argue that counter-history 'opposes itself not only to dominant narratives but also to prevailing modes of historical thought and research methods'. Therefore, we relied predominantly on storytelling as an African research methodology and epistemology (Mucina 2011a). Our primary research method was biographical narratives, which allows the narrator to choose what to narrate without being guided by the researcher (Willemse 2014, p. 39).

We continuously held biographical interviews with two women, our mothers, who were the main research participants. The first author's mother, Alice, a child of a single, blind young mother, was born in late colonial Kenya in the Akamba community of Kitui. Alice's mother's narrative of stoicism, narrated over the years, and her grandmother's will to overcome the gendered, generational, ableist, and cultural discrimination in her context inspired this research. This resilience and resistance were also read through the emergent agency of her mother's grandmother, who defied tradition, refusing to marry off her blind daughter. This determination by her grandmother revealed essential aspects of gendering in precolonial Africa, where senior women held power. In moving beyond the role of unbridled male power (a colonial category and concept) in the processes of growing up, such agency was not just based on gender but also seniority.

The second author's mother, Minji, is from the Luo community in Siaya, and was the daughter of a young mother who resisted and ultimately endured her arranged marriage to a polygamous older man. Coming of age in late colonial Kenya, Minji withstood daughter discrimination when she was withdrawn from school in class four while her brothers went on to complete high school. Her resilience is seen in her fortitude in transforming the gendered violence and prejudices she faced due to her identity, including as a mother of five daughters,[1] into a life's work of advocacy and activism on gender equity in education and youth wellbeing.

In order to strengthen the perspectives that emerged from the biological narratives of these two women, we also repeated life history interviews with several other women in these two communities. These women were born and/or came of age in the late colonial period in Kenya (1940–1963). This category was broad, meaning that different generations participated in the research, with births ranging from the 1940s, 1950s, and early 1960s. We also interviewed older women and men in their late 80s and 90s who had interacted with or were contemporaries of the two main research participants. The intergenerational nature of the inquiry meant that some women shared perspectives of adolescence from what they had learned from their mothers or grandmothers, not just their lived experiences. There were also intergenerational exchanges when we held focused group discussions with these women of diverse age. Such inter- and cross-generational narratives have implications for contextualising the shifts and contradictions in perspectives of coming of age. This is because, in some cases, women of diverse ages held different perspectives. Therefore, while seeking corroboration of collective views through exchanges between generations, we were also aware that the processes and practices of growing up are distinct in time and place. The perspectives that emerged from these discussions, as we present, also acknowledge that memories are continuously being reshaped through the processes of narration (Mucina 2011a; Willemse 2007).

Even though we relied on the age of our study participants, we note that most of the older research participants did not know their precise year of birth or that of their older siblings, whom they also referenced. They relied on recollections from relatives and significant time markers, including famines, seasons of plenty and want, epidemics, and the advent of key changes in the colony, like the use of coins, to estimate their age. Most of the women and a few older male study participants had not received formal schooling, and some only attended a few years of the lower classes. While some women noted their agency in refusing school because they did not see the value of formal schooling, for some, gender discrimination in sending girls to school pushed them out of school or influenced their disengagement.

Inspired by Marker's (2003) formulation that experience exists in spaces and places, we also observed and allowed the women to re-enact their spaces, places, and sites of growing up. This involved visiting former schools, ruins of childhood homesteads where they existed among the Kamba, and the traditional spaces for guidance and counselling among the Luo, called *siwindhe*. As is characteristic of African orature, the narrations included body and corporeal performances and movement. Through their narration, study participants articulated the smells, sounds, sights, and feelings from memories of their past, drawing us into such forms of being and belonging (Mucina 2011a, p. 6; Rozmarin 2021, pp. 34–35). For example, we often observed them burst into song and even dance when reference was made to a popular song as they reminisced about their coming-of-age rituals.

Since dancing ceremonies comprised some of the spaces in the process of growing up, we held a dance and song workshop in the Akamba research site, where participants donned traditional attire and re-enacted the dancing spaces. The workshop enabled us to observe body movement and sounds in order to understand women's relationships with the past, place, and ritual practices. This workshop was also an aesthetic space that communicated the vitality of life through adolescence (Irvinson and Renold 2021). For one man, the workshop was a site for reminiscing on the loss of the past. He rationalised his inability to dance well, arguing that he had wasted his time in formal colonial schooling with nothing to show for it in the present.

After obtaining oral consent from the study participants, all discussions were audio recorded and translated into English from Kamba and Luo by the two research assistants, who spoke the languages of the two communities. We also organised a radio call-in programme at each of the research sites. These programmes allowed the broader communities (to which these stories belong) to engage with the stories before they were published and viewed by our readers. Post-broadcast focus group discussions with the study participants helped follow-up on, clarify, and supplement issues raised during the radio shows.

In terms of ethics, despite their advanced age, some participants were concerned with the daily chores like fetching water and tending farms, sometimes ostensibly performing the identity of a hard-working adolescent girl of the past. We were also aware of the messy corporeal ways in which our bodies were simultaneously implicated as insiders or outsiders, and how our experiences disaffirm those of the participants (see also Irvinson and Renold 2021, p. 69). For instance, we vicariously connected with the pain of the participants' ageing and sick bodies, including those that had endured a lifetime of pain. In becoming with such bodies and for care, we provided participants with snacks, offered transport refunds, and organised transport in cases where focus group discussions were organised in a central place. We were also sensitive to overtaxing participants' time and comfort, and made centrally organised meeting brief. We were also reflexive of the spaces and loci in the global North where we are professionally located. This positionality means that we might be seen as part of the unequal ways of knowing that we are trying to engage with in this research (Turner and Norwood 2013). Therefore, we maintain a critical awareness of this positionality, even as we endeavour to represent the research participants in respectful ways.

Memory as a presence was a form of healing for some of our research participants, who noted that this study was a space for easing stress (see Williams and Molebatsi 2019). However, memories for the marginalised are not always beautiful. We show how, for some, there was re-membered pain, including the pain of infertility. As we later show, this pain also became our pain as we carried out research with women and men who were confronting mortality (see also St. Pierre 1997). As the first author noted in her field notes, 'How does researching and writing about participants, who while telling their stories, for posterity, are confronting their mortality affect researchers? Those who keep saying 'we are going', amidst the laughter and the hushing, not yet'. We, therefore, research but also write as we process the loss and the trauma of death. We say with the Siaya study participants that death is inevitable, and it is *dalaji te*, 'the home of everyone' (Ngutuku 2022).

Participants also wanted to recognise themselves in their own words in their stories and names, so we endeavoured to use their names where specified. In desiring accurate representation, some participants frequently sought to verify if we were in sync with their narratives (Mucina 2011a). For example, when interviewing one male participant in the Luo research site, he stopped the interview each time his wife stepped out of the house because he wanted her to corroborate his account. He also insisted that the first author, seen as a linguistic outsider then, must capture the narratives correctly. Similarly, Alice, one of the two main research participants, continuously narrated herself by asking the researcher, '*Niwakwatya?*'—'Are you getting that?' We have, therefore, endeavoured to respect this epistemic agency.

Being reflexive about our interpretations meant that we constantly sought out perspectives and interpretations from other people. In doing so, we were inspired by the research done by St. Pierre (1997) among women in her mother's community in Ireland. She argues that we ought to bring the outside to our analysis during research, or what she sees as knowledge by others, or response data (St. Pierre 1997, p. 184). Checking these views with others was also inspired by the view that, like stories told by the fireside, this story of coming of age does not belong to us or the women alone, but also to interpretations by others (Achebe 1988, as cited in Mucina 2011b, p. 51). Our quest for response data should not be read as discrediting the women's voices. For example, when the first author wanted to check Alice's interpretation of a proverb with Alice's husband, she asserted, 'Even if you ask anyone else, that's my truth.' By assuming that the author was trying to discount her truth using the truth of her husband, she was resisting the hierarchies of credibility that characterise research on indigenous knowledge. As noted by Debele (2021), we must not be held captive by the same structures of epistemic violence we strive to engage in this paper.

Even though we see ourselves as cultural insiders, deeply invested in this project, we are also aware of the perils of what is seen as 'cultural insiderism' or a feeling that our interpretations are better off because we are cultural insiders, and that those from the outside are wrong (Debele 2021, p. 100). We are, therefore, guarded in our arguments as we continue to be reflexive about our power as researchers. In so doing, we draw from the Nigerian-born poet and novelist Okri (1997, p. 43), who encourages those like us who 'tell from the inside' to be reflexive of the fact that 'storytelling is always, quietly, subversive. It is a double-headed axe. You think it faces only one way, but it also faces you. You think it cuts only in one direction, but it also cuts you.'

In the subsequent sections of this paper, we draw from a few illustrative examples to present the practices of growing up.

## 5. Notions of Coming of Age

This research revealed no distinct stage known as adolescence, but rather, the different rites, expectations, and responsibilities that marked growing up and transitioning into adulthood. Even though the study participants attempted to translate this process using normative age-based categories, age sets other than precise age were used to demarcate or represent coming of age. For example, among the Kamba, after childhood, *Twana*, they moved to the stage of *Kavisi/Kelitu*, where one was not seen as a child but was not fully developed either. This would correspond with pre-pubescence. The next stage was '*Kelitu* or *Kamwana ka mweewa*', a stage not just marked by the beginning of physical changes like breast growth and first periods for girls, and facial hair and deeper voices in boys, but as the beginning of the stage for participation in dancing ceremonies. The nocturnal dances were the spaces of coming of age where girls and boys gathered in the village grounds to dance. These boys and girls were also referred to as those who 'carried clothes for the real dancers', because they would be on the side-lines, watching the older cohort *kalani* or *nzungi*, who participated in dancing events. While there was a stage known as *mwiitu/mwanake* for girls and boys, respectively, this was more about societal expectations of behaviour and responsibility. The very mature girls were known as *Ngulumbu* or *Ngilituki*, articulated in an onomatopoeic way to emphasise that they were 'full' and thus, ready for marriage.

Participants noted that girls in this age set were also known as *Mbinga Kivalo*, or those that closed the dancing ground, indicating their maturity. Focus group discussions revealed that marriage among the Akamba did not necessarily define maturity. The newly married women would be matched with mentors who were older women in the community. We also learned that in some cases, young, newly married women would not be sent back to their parents if they made mistakes until they had given birth to five or so children because they were seen as young and immature.

Among the Luo, participants noted that a very young child was *nyathi* (*ma wuoyi*) boy and (*ma nyako*) girl, and they typically received education from their mothers and slept in the same hut as their parents. Pre-adolescent boys and girls (approximately 7- to 14-year-olds) relocated to a designated communal sleeping and learning space called *Siwindhe*. This hut belonged to an older often menopausal woman called a *dayo*, who was charged with educating young people on various subjects. Coming of age for adolescents, referred to as *rawera* (*ma wuoyi*) male and rawera (*ma nyako*) female, were marked by physical changes including facial hair and deeper voice for males, and breast growth and menstruation for females. At this stage (around 14), male adolescents moved from the *Siwindhe* to sleep in the *Simba* (bachelor huts), where they learned about courtship, dances, hunting, and wrestling from older unmarried males.[2] The *simba* was also the space where both male and female youth engaged in *chode*—dating typically done in groups where they could engage in heavy petting but were strictly cautioned to refrain from penetrative sex. Additionally, young men received more advanced education from older men in the male hut called *Abila*. Adolescent girls stayed on in the *Siwindhe* in preparation for marriage, receiving more advanced education on tradition, history, and values, as well as sexual education.

A fully developed girl was deemed full (*pong'*) and thus, ready for marriage, termed *dhi tedo*—literally going to cook. Indeed, participants noted that a Luo girl's worth was seen in her marriageability and motherhood, which were significant social markers for attaining adulthood and conforming to societal norms. Those who delayed or failed to 'get a cooking place' earned derogatory descriptors like *diwo*, 'left behind', denoting their failure to attract a suitor, and thus, their 'on the shelf' status. An even more pejorative term, *Odhi oduogo*—'the one who went and came back'—was used for a female who came home, ostensibly after a failed marriage.

## 6. Guidance and Responsibilisation: Being with the World and Others

In both contexts, coming of age was tied to specific initiation ceremonies. During this stage, young people received guidance and instruction on various issues, including marriage, culture, relationships with other people and the environment, the spiritual world, the land, animals, and the future. Among the Akamba, the first initiation ceremony, which involved circumcision for both boys and girls, included some form of counselling. The subsequent second initiation ceremony (*Nzaiko ya Mulili*) marked their official coming of age. This ceremony was carried out in the 'month of the knife' (*Muvyu*), or what is now known as October, marked by the rainy season after the ceremony. During the second initiation ceremony, girls and boys had mentors—*Avwikii* or *ataithya asingi*—who guided them on proper behaviour and sexuality. Males were also instructed on the community's secrets (Jacobs 1961). They performed various rites, like *Kwatha Nzavili*, shooting an imaginary elephant, or proving their manhood or strength. Other activities included hunting and learning about the ancestors. Girls were also responsibilised in various ways, including learning essential skills linked to their caregiving and community management roles, as well as sexual education.

Among the Luo, initiation involved *Nak*—the extraction of the six lower front teeth for both males and females—as a critical marker of coming of age. Not only did this initiation ritual test the courage and endurance of each group, but it also helped to bond the cohort and identify one as a Luo. It also had the practical purpose of enabling medicine administration for diseases such as lockjaw. Participants noted the considerable waning of this practice by the late colonial period, pointing to the dynamism of cultural practices.

In both contexts, girls received guidance and had conversations to support their sexual maturation, including how to manage menstruation. For example, they were cautioned against the careless disposal of menstrual products, couched in the taboo that ill-intentioned people could use these to make them infertile. The norms around initiation can be interpreted as connecting with the need to conserve the environment or to be at peace with nature. Konyana and Konyana's (2021) work on the role of menstrual taboos among the Ndau women in Zimbabwe reveals that such taboos cultivate positive values around the conservation of the environment. The practices around adolescence were also connected to the land and other non-human beings. For example, the study participants noted that among the Kamba, during the second initiation ceremony, there was singing and dancing in a calculated posture known as *Kutomba* as the initiates went home after the ceremony (Jacobs 1961). This also involved singing around objects or people they encountered on the way, as noted by one participant, 'if we encountered anything, human or non-human, we would stop and *kutomba* around it'. This included fire, animals, a road junction, a river, a bridge, a woman, etc. This calculated posture of walking and singing was also interpreted as a considered approach to life that characterises African relationalities. Such practices support the ubuntu relationality that there is energy in everything, including nature, humans, and other things Mucina (2011a, p. 4). Reinspecting the natural world shows the special relationship between nature and identity and belonging in Africa. Indeed, being, as a form of belonging during these ceremonies, was not just what Molefe (2018, p. 9) calls 'being-with-others', but was also being with the environment, places, and the spiritual world.

Beyond initiation ceremonies, guidance was continuous through interaction with diverse people and extended family, as well as the designated spaces for young people, including being responsibilised. Luo socialisation/education was completed within traditional educational structures. For male youth, this was both in the *simba*—bachelor hut—and within a special hut, *duol/abila*, in each homestead, where the male head of the homestead held court, met, and conferred with his peers, and where older males mentored young men. In the *siwindhe*, female youth received advanced education on various topics linked to their caring roles, including sexuality education, bodily autonomy, care, and comportment in preparation for marriage. Participants noted that married girls who visited their homes also joined in these lessons in order to impart knowledge on their experience in marriage. The significant role of the *siwindhe* as a forum for the education of young people is encapsulated in a Luo saying that was used to chastise those who had a poor grasp of the socio-cultural norms, as noted by one of the study participants:

*"Iming nadi ka ng'at manene ok onindo e swindhe!"* (You are ill-educated/ill-informed like a person who did not sleep in the *Siwindhe*!)

Among the Kamba, girls were guided by their mothers, aunts, and grandmothers, and in other everyday spaces, like the communal harvesting of food where several women gathered to help each other and perform daily chores. Boys were guided by their fathers and grandfathers in *Thome*, the outer court in a homestead. Traditionally, *thome* was a space where men sat by the fire, sometimes for the whole day, ready to defend women and children in case of any attack. Among the Luo, the education and guidance provided by the *Dayo* in the *Siwindhe* included sexual education, or what the women in our research framed as 'how to take care of themselves'.

In both communities, therefore, there was a clear recognition of the importance of sexual education as a critical component of these learning sessions, typically held between dinner and bedtime each day. Recognition of young people as sexual beings undergirded these structured mechanisms of pubertal and adolescent education and guidance. Such guidance also shaped and supported their sexuality, burgeoning sexual maturation, self-care, and transition into a young woman or manhood and, ultimately, marriage.

### 7. Gendered Notions of Sexuality: Engaging Narratives of Male Permissiveness

One of the key themes in the women's accounts of growing up was gendered notions of sexuality. This is also an area that has received a lot of commentary in the literature on girlhood, contrasting the controlled sexuality of girls with the pressure on boys to demonstrate virility and fertility. Our research affirmed these perspectives around the permissive sexuality of male youth, as noted during a focus group discussion:

> 'Young men did not have a lot of rules governing them, and there were even avenues for their sexual exploration, including with widows.'

A young man was not supposed to marry a girl with a child born out of wedlock. In contrast, a young woman had to be *silili* 'pure' and exhibit proof of her virginity during the first conjugal act of marriage—the deflowering, *ng'ado ringre*. Two designated female contemporary *nyieke* (women married to the husband's brothers/cousins), called *Jondaria*, served as witnesses to this act and would hand over the stained beddings as proof of the bride's virginity.

Dominant sexual norms and expectations in traditional societies were often codified through proverbs, seen as bearers of important indigenous knowledge. For example, notions of permissive sexuality for males were codified through proverbs, revealing the expectation that men should display macho sexuality ([Hussein 2005](); [Mfecane 2018]()). For example, among the Akamba, one proverb presented young men as hyenas who must get out of their lairs at night to hunt prey. At the same time, another saying represented the sexuality of women as a forest that could be explored. Yet another proverb depicted young men as buffalos who must always break in through the rear of the cattle pen since the gate is fortified. This contrasts sharply with girls, whom one participant likened to cows that must be controlled. In both communities, boys were allowed to stay out late compared to girls, and their mothers would keep food for them in the bachelor huts.

Among the Luo, girls were expected to maintain their chastity in order to raise good families, and were strictly cautioned to avoid pre-marital pregnancy, which was highly stigmatised. Some women noted that they were expected to conform to these norms since a girl's personhood before and after marriage reflected on the larger honour of the family and clan. Girls who got pregnant before marriage, derogatorily referred to as *ich simba*—pregnancy of the bachelor hut—would be married off to an old man (see also, [Ngutuku 2023]()). The paternity of the child was unmasked during delivery, as revealed during an FGD.

The responsible (young) man would be known during the delivery as the midwife, and the women helping with the birth would compel the girl, under duress of labour pains, to name the child's father for the baby to come out. There were threats that the baby would not come out if she mentioned the wrong name.

Despite the strict codes around their sexuality, girls did not accept unquestioningly the explanations around the consequences of unbridled sexuality. For example, Ndolo, in her late 90s, who was unable to have children for many years after marriage, questioned the view that infertility was caused by promiscuity, noting:

> 'I was a very obedient child; I did not go out indiscriminately with boys. I wonder why I was not able to get a child.'

Traditional codes and discourses about proper sexual conduct were thus not always in line with what young people wanted or practised (see also the research by Kamwendo [Naphambo 2021](), in Malawi). The fact that some women questioned these rules contradicts [Mbiti]()'s ([1986](), p. 312) view that in African societies, 'individuals automatically accept religious beliefs without questioning them'.

There was, however, no unanimity in the two cultures on perceptions of premarital pregnancy. These perceptions were different based on the narrator's age, indicative of the dynamism and how what we see as tradition (*Chik* and *Kithio*) among the Luo and Kamba, respectively, also shifts over time. For example, among the Akamba, there was a view that pregnancy affirmed a woman's fertility and those who got pregnant before marriage were saved from the rituals around infertility. There was a saying that God had done their

*ngondu* [rituals]. Syokau, who says that during the great famine of 1928 (*Kakuti*) she was about four years old, noted:

> 'A woman would still be married with her grown-up sons, or what she called sons who came carrying bows and arrows, but things changed over time, and some people started seeing it as a problem.'

Thomas (2005) has expressed the same view in her research among the Ameru of Kenya, where premarital pregnancies proved a woman's fertility. In both communities, a man responsible for the pregnancy was expected to compensate the girl's father. Later, colonial laws mandated that the Pregnancy Compensation Act governed pre-initiation pregnancies, and the man responsible compensated the girl's father for making his daughter pregnant. In late colonial Kenya, the Affiliation Act ensured support for the children of single mothers (Kioli et al. 2012; Ngutuku 2006; Thomas 2005, p. 106).

Our research reveals that male permissiveness was, therefore, not the only way of being around sexuality in these two communities, and there were diverse ways in which men were also held to account for their actions. For example, while young men would have several girlfriends among the Kamba, they were not supposed to have sexual relations with the girl they were planning to marry, or what is seen as notions of relative chastity (Maithya 2002). Discussions revealed that the groom would be asked if he had 'explored the forest' during the marriage negotiations and ceremony. If he answered in the affirmative, he would be given a half-full calabash of the wedding brew, which would be an embarrassing putdown. Participating women also affirmed their sexual agency, often absenced in dominant accounts, noting that girls were allowed to keep lovers even when marriage negotiations with another man were ongoing. Further, during the dancing events, *wathi*, the younger married men, watched over the young people to ensure that young men did not take advantage of the women (Maithya 2002).

In both contexts, there were other ways in which girls exercised agency in asserting their sexuality. This agency is often absent in what Arnfred (2015, p. 152) sees as the panoptical gaze (or surveillance) of Christian colonialism, which did not have a place for female sexual agency. This, as he argues, was because seeing women as agentic was frightening to the colonial power that positioned itself as protectors of women against patriarchy. For example, among the Akamba, women noted how they would tell their lovers which side of the bed they slept on, and the boys would insert a stick through the grass wall to signal them to come out at night. Two women in their late 80s noted how they would sneak out by opening the makeshift doors of their house and return very early in the morning.

Among the Luo, as part of becoming adolescents in the *siwindhe*, some women used the power of the *Dayo*, the elderly gatekeepers of girls' sexuality, to call the gender and sexual order into question. For example, some girls organised with the *Dayo* to sneak out at night and see their lovers. In the late colonial period, this involved the boys giving the *Dayo* some incentive in the form of coins or tobacco for her pipe (*kwesi*) in order to buy freedom for the girls at night. However, the boys in question had to be known by the *Dayo*, as noted by one participant:

> 'A mature girl would arrange to visit a mature boy from another clan, and *Dayo* in the *siwindhe* would cover up for her. When he came for the girl, The boy would give *Dayo* money to buy some tobacco *for her pipe (kwesi)*. The *Dayo* would open the door as early as 5 am so the girl would enter the *siwindhe*.'

Therefore, these girls resisted control not through opposition, but through entanglement with the power of tradition and gender power (see MacLeavy et al. 2021, p. 1559).

Countering men's permissive sexuality in some contexts meant lying to boys that one was on her menses or bereaved when they met with boys during night dances. For example, among the Akamba, people were supposed to abstain from sex during the grieving period (*Ivou*). Some girls also relied on information about the boys they were in a relationship with to determine if there would be repercussions for engaging in sexual conduct. For example,

some men would reportedly be taken to herbalists by their fathers in order to prevent them from making girls pregnant before marriage. One of the Akamba study participants, Wanzoo, revealed her awareness of what her lover's father had done that assured her that she would not become pregnant.

Cultural practices and gendered norms powerfully prescribe and endorse behaviour. This includes young people's sexual behaviour and prevailing beliefs around men's permissive sexual desires, which perpetuate sexual risk-taking among men versus young females'-controlled sexuality. However, our analysis reveals much more nuanced perspectives. Rather than passively conforming to these set ideals, girls resisted in surprising ways. We must acknowledge such resistance as part of their being and becoming adolescents in these spaces.

## 8. Marriage Practices

In this section, we focus on marriage as one of the processes of coming of age in these contexts. In addition to arranged marriages, we also explore the practice of marriage by abduction, which has been a subject of commentary by several researchers, locating how it was experienced by the women in our study (Jacobs 1961; Maithya 2002; Evans-Pritchard 1950).

We start with Alice, one of the two main research participants, who had a suitor whom she loved. As she noted, since she was a Christian, she was planning to get married through a church wedding. Her grandparents, however, betrothed her to her grandmother's kinsman. Alice's inaction in her becoming was pragmatic, and she rationalised it through her life circumstances since the potential groom had already paid part of the bride's wealth. Alice was also concerned about her mother's respectability and did not want her poor, blind mother to suffer if forced to refund the wealth, as was common practice. She noted, 'I feared that the community would mock my blind mother for accepting bride price and this would be considered as scamming the man'. Therefore, Alice capitulated to the dominant norms around marriage by justifying her willingness to get married, not as a result of being forced, but as fulfilling her roles as her mother's caregiver. Therefore, Alice's inaction helps us see that resistance does not always mean escaping power relations, but being entangled with them.

Similarly, some Luo women noted the difficulties in resisting marriage when the bride wealth process (*nyombo*) had already commenced:

'If the bride price payment had started, there would be no turning back. It was challenging to change your mind because of respect for your father. It would be wrong if your father were made to release the cows, which he happily accepted. The old men had all their hopes on these cows.'

Alice's husband expressed a different perspective, arguing that he wanted to help his blind mother-in-law, thus presenting himself as the angel that his mother-in-law did not have:

'I had been told that blind people were on the streets begging, but when I went to the streets, I did not find her. Most of the time, she was left home alone. I started supporting her and could not stop for anything. [We] had given bride wealth, yes, but I also liked her daughter.'

For Wanzoo, in her 90s, her path to and in marriage was complex. She narrated that she had a lover who was a soloist at the dancing events. However, like most young men in colonial Kenya, he left for Mombasa, and she did not hear from him for three months. During a major famine in her community, Wanzoo accompanied her family to a neighbouring Kikuyu community in search of food. A neighbour gave the family thirty Kenyan shillings (2 U.S. dollars by then) for transport. Later, when Wanzoo's family wanted to pay the money back, the creditor wanted Wanzoo as a wife for their son, who was working in Tanzania, as she noted:

'When my stepmother wanted to pay back, his [husband's] father declined and said they only wanted what my mother went with to Kikuyuland. I realized I

was the one they were referring to. Things happened so fast. I accepted because this family had helped us a lot, and I would not have done them justice if I had refused to go.'

In understanding Wanzoo's narrative, we must locate it in colonial modes of accumulation, which, for example, involved incorporating Africans into the world's economic systems of using money. In addition, the erosion of stable traditional food systems and storage practices interfered with food production and led to frequent famines (see Muthui 2021 for research on women and famine in Kitui, Kenya). During such famines, young women were married to wealthy families who paid in the form of money, not cattle, as was previously the case. However, what we intend to emphasise here is Wanzoo's agency. Her actions should not just be read as acquiescence to injustice, she also drew from the power of tradition in her seeming inaction. For example, her stepmother advised her that she could always return home if the marriage did not work out. Jacobs (1961) has reported similar grounds for separation or divorce, including if the man was mistreating the woman and if he had a tiny piece of land. Another reason was if the woman was not bearing children, and the man was unwilling to take her to a medicine man. Similarly, among the Luo, participants noted that if one discovered that the husband was cruel or a wizard (*Jajuok*), the woman could return to her parent's home, even if the bride price had been paid.

Like Wanzoo and others, gender power and other forms of female networks also came to bear on their decisions to accept or reject arranged marriages or endure them. For example, one of Wanzoo's stepmothers, a product of the same type of marriage, also encouraged her to return to her marital home, encouraging her that one day she would have a nice family of her own. For several of the women whose marriages were arranged this way, such hope was not just an emotion, but a sociality of resilience and a temporality of a future yet to come made them overcome the difficulties of what would be seen as a forced marriage. In their sense of becoming, they lived intensively in the present, or what we would call tenacity, and overcame or engaged with difficult moments whilst hoping for a change in their situation in the future (Ngutuku 2022). Such hope was not a cruel form of optimism, as we shall see later, but an end in itself.

Staying with Wanzoo, her agency was also obtained from the fact that she still tried to negotiate on when to get married and, therefore, get some leverage around the process:

'I told my man to come the following day, but he could not hear any of this. I wanted time to say goodbye to my lover and get one of my closest girlfriends to escort me. I had selected my sister-in-law, who had been sent away from our home because she could not bear children.'

We further see her agency when Wanzoo decided to return to her village after her husband failed to come back to the village after marrying her, as she noted:

'I had no physical husband, and I wasn't even allowed to be in any relationship with any boy. I said to myself that it was even better when I was a girl because I had my lover.'

This was not uncommon in the colonial period, when young men got married and immediately migrated to urban areas to seek work, leaving their wives behind. This was also the fate of Ngambi, in her late 80s, whose second husband was absent during her marriage, 'I did not even know my husband; his brother and mother came for me.'

When Wanzoo's husband's family got wind of her impending re-marriage, they foiled the marriage negotiations, forcing her to stay, again putting limits to what agency was possible, at least in the meantime, as we will return to later.

Here, we briefly return to our argument that resistance does not always call for celebration and can also be bound up with pain and suffering (MacLeavy et al. 2021). For example, in the case of Esther: when other forms of resistance, like refusal of household chores and putting a lot of chillies in her potential groom's food, failed, she attempted to end her life as she noted:

'My parents had arranged for me to marry a man from a wealthy family, but I did not love him. He came one day, and my parents told me to cook for him. I put a lot of chillies in his food to discourage him, but he did not eat the food. I became rebellious and refused to perform my daily chore of scaring birds from our farm. When this did not work, I attempted suicide. I woke up from my deathbed two days later. The man's parents backed off, and I married the man of my choice.'

Esther's account confirms the research by Jacobs (1961) in colonial Kenya, which showed how girls used the threat of suicide if forced to marry a man they did not like.

Further, in both contexts, while girlhood and womanhood were entangled in terms of unequal power relations, gender power was applied more in terms of seniority. The older women exercised power over younger women, as revealed during an FGD with participants among the Akamba"

'If one resisted and eloped with another man, the mother would come for them and wash off the oil which had been applied.'

The participant above references the marriage ritual of applying ghee to a newly married woman in a ceremony known as *Kuvikwa*. This social and religious ceremony lasted for seven days. Women also reported that such excesses of agency, like getting married without authority from the parent, were met with the possibility of a curse by the mother, implying potential infertility. Here, we see how specific women, like mothers, were also powerful. Macharia (2012) has argued that among the Yoruba, the introduction of gendered colonial categories failed to consider women who were also powerful this way. These accounts where women are either cursed or 'de-oiled' when they make choices may show the impossibility of agency for such women. However, we must also acknowledge these incongruent forms of agency.

In some contexts, if the bride wealth had been paid and a girl's family did not want to release her, then she would be taken by force or captured. This was known as *ywacho* among the Luo and *Kuvulwa* among the Kamba. Several women in our study had been married this way. This practice was, however, remembered differently by different women, also based on their age. Among the Luo, one version revealed that it was performative, with the process starting before *ywacho*, when some 'spies' would be sent to check where the girl slept, and she would be taken at night, sometimes after breaking the makeshift door to the house. The girl's sisters would put up a mock rescue, but they would not stop their sister from going. Some participants noted that it was taboo once the husband's group had held her arm for her to be taken back. Others revealed that the practice was performed by the brothers in the two families as a way of showing their strength in capturing their sister-in-law or defending their sister. If the boys' team lost, then the girl would not go to his home on that day:

'If you didn't want to go to the boy's home even after he had finished paying the bride wealth, some boys would be sent to take you. Someone would visit you and trick you into escorting them. You would then be taken along the way.'

Like in the case of Alice, our research revealed that it was not just age and gendered differentiation that defined young female subjects and the associated processes of growing up. Other life forces in their contexts undercut how growing up was experienced (Hynd 2021; Loew 2012). For example, for Ndolo, these factors included being orphaned, her father's disability and migration to urban areas by men. For instance, Ndolo's family had betrothed her to a young man working in Nairobi. This was the case for many young men who had to leave the rural areas for urban centres in search of jobs during the colonial era. Ndolo's husband was, therefore, expected to return to Nairobi for work, making Ndolo's capture inevitable. She narrated how she was dragged along the path from their home to the prospective groom's house, reminiscing about the pain:

'They came (the groom and his brothers) and told me I had to leave. . ... I refused, but they could not hear any of it. I was dragged on the path. . .They flogged me like a donkey into submission.'

She narrated while placing her hand on one cheek and sometimes demonstrating the flogging, the pain etched in her eyes, almost 70 years later.

'My mother had died when I was only two years old, and my father was deaf. As an only child, no one was there to protect me.'

Ndolo's experience does not resonate with other participants among the Luo, who noted that an orphan child (*kich*) would not be captured. We also need to understand Ndolo's predicament from the fact that all her agemates were already married, and as she noted, 'Her breasts had become cold', meaning she was late for marriage. She had, earlier in our discussions, self-fashioned as an agentic subject through hysterical laughter while boasting of her strength as she resisted a previously arranged marriage by fighting off the suitor (see Emmerson 2017, for a perspective on laughter's affective and agentic role). She noted:

'[He] (the potential groom) came and slept on my bed frame, which was made of hide and skin and supported by some poles. I waited until he was asleep and removed the poles. He came tumbling down; sheep and goats were under the bed[laughter]. So you can imagine how he felt; I never saw him again.'

Like Ndolo, 100-year-old Anyango from Gem shows that it was not just tradition but also the girl's muscular power:

'I experienced *ywacho* at my grandmother's place. When I left my grandmother to go back home, I met this group of boys, and they stopped me. I got scared and started fighting them and defeated them. Only one boy subdued me because he was stronger than all of them. I was taken to a home that I did not know, and I spent the night there. There were many cows in that home.'

The story of the acquiescence of another participant, Akoth, was complex and revealed how girls' understanding of sexuality played out in this practice:

'I came to visit my cousin, who was married to Oloo. Oloo locked the three of us in his house and asked one of his friends to choose one of us as his wife. The friend pointed at me. When it was time to leave, Oloo followed us together with some of his friends, captured me and locked me in the house. The next day, he sent word to my father that he had me in his house and was ready to pay the bride price for me. My father was angry and ordered him to let me return home immediately. Oloo then went to Muganda—the chief in my area- and sent him to talk to my father. Muganda took some cows to my father, but he refused, saying he wanted to see me. Muganda went back to my father again without the cows. My father refused, and I was set free. My father asked me if I wanted to be Oloo's wife. I told him, 'If I had not wanted, I would have escaped when I was at his house'. My father saw that I was ready to be Oloo's wife and allowed him to start paying the bride price. Oloo was a mature man, and he was handsome. He was not a boy. One day, he sent some men who came and took me by force from my father's home. They took me from the kitchen.'

In this narrative, we emphasise not only the process but also Akoth's subjectivity and positioning, and her acceptance to be a third wife that reflected the sexual norms of her time, which we should not judge in the present. Here, we also pause to clarify that even for some marriages, including those seen as 'wife capture', where the women are represented as having no agency, there was background work to vet the groom or bride and their family. Among the Luo, the boy asked a sister or aunt to be *ja wang' yo*—a go-between—when looking for a wife. *Ja wang' yo* helped to eliminate the chances of marriage to a family with questionable values by ensuring that the groom's family passed muster:

'A boy would tell his father he wanted to get married, and the parent would send him to an aunt or elder sister to help him look for a girl. After the aunt had identified the girl, the boy would talk to the girl's father and report to his

father. The father would then send word to the girl's home, requesting a meeting to discuss marriage.'

The vetting of the girl by *Ja Wang'-yo* would include looking into her character and how responsible she was, including whether she would tend a farm and manage a home with enough food in the granary. Calsina, in her early 90s, noted in a discussion with Minji and 87-year-old Atieno that the girl who exhibited the capacity to do these chores was seen as a responsible woman and a good choice for marriage. Among the Akamba, they would also check if the girl came from a problematic home (*musyi wa kitai*) or if the man had an older sister whose bridewealth could be used to pay the bridewealth of the prospective wife. In these communities, therefore, girls were not just commodities for sale.

## 9. Reading the Future That Has Come, Not As-Yet, and Has Gone

In this section, we draw from the lived experience of a few research participants in order to provide a perspective on their imaginaries of resilience as hope over time. We read about this resilience from their past, the future imagined then, and their life during and after this study. We are interested in how the temporalities of waiting for the future have turned out in their present (Ngutuku 2022). For some of these women, the futuristic girlhood/womanhood in the past is a continuing present. In this way, resilience emerges as a lifetime way of being and becoming with their world, here and after.

We start with Ndolo, whose painful capture for marriage we presented earlier. After battling infertility for several years, she finally had two daughters. In order to maintain her name, or what was seen as ensuring that the central pole in the traditional granary in her house was not broken, she sought marriage to another woman in a ceremony among the Akamba known as *Iweto*. This is a symbolic woman-to-woman marriage. This is an institution where a woman who did not bear children can get a 'wife' who bears children on her behalf. In some contexts, the woman is referenced as a wife or a daughter-in-law. While Ndolo's *Iweto*[3] has bipolar disorder, she is happy that she now has a name through her grandchildren, who sometimes support her. She, however, relies primarily on village members for support, and has refused to go and live with her only daughter in a faraway community. Doing so would be leaving behind the land and place of her once-prolonged adolescence, a land of fulfilled and unfulfilled dreams. However, she finds meaning in how she helped people when she was younger, including mentoring newly married women. She thinks these efforts were repaid when a neighbour helped construct a flush toilet in order to overcome the problem of her arthritic achy knees.

Similarly, Ruth, in her 80s, is happy. In her early days of marriage, Ruth not only helped get another wife for her husband, but also got an *iweto* (symbolic wife) since she could not bear her own children. While both Ruth and her husband confess their love for each other, Ruth feels like she is half-human since she did not have a child as required in the society of her time, then and now. In tears, she blames her uncle. Her uncle had vowed that none of her daughters should get married and that if anyone defied, they would suffer. She is still asking the question, what is life without a child? Malinda, who did not bear children of her own, was seen as the icon of village female agency when, as a young woman, she fled an abusive marriage and remarried. She is praised for singlehandedly repaying her abusive husband the bride price. During this study, at almost 100 years old, she was endearingly supported by her *iweto*, proudly showing us what a lifetime of resilience looks like. She passed a few months after this study, leaving several grandchildren behind.

Alice, whose marriage was arranged, looks back at the challenges of growing up and finds joy in her supportive family and grandchildren, who keep her company. She veers off-topic, and, through banter, tells how she regrets why she got married into the family of the wealthiest man in the village (her husband's father, who owned more than 100 heads of cattle) but whose children lived in poverty. During our conversations and as we visit her spaces of childhood, she looks at the guava tree that marks her mother's grave, who has been dead for over 40 years. She wishes her blind mother was alive to see how things have

turned out well for her. Even though the circumstances of her life denied her education, she positions herself as the pillar of her family.

However, retrospectively, Alice still longs to belong to the girlhood of her time. This makes us take a flight back in time to contextualise this longing in our telling. For example, Alice and other participants had characterised the spaces of growing up, like the dancing ceremonies, as spaces for 'feeling happy' and showcasing beautiful youthhood by wearing elaborate jewellery bought by their fathers. Culturally, adolescence was also seen as a space for being responsibilised and learning to perform household chores. However, for Alice, a child of a single, blind mother, things played out differently. For example, she did not have to wait until the initiation stage to learn to look for firewood. As a first-born daughter, she had learned to carry heavy containers of water early, sometimes with her blind mother supporting her neck from behind. While others were going for evening dances or later to shows organised by the colonial government, sometimes she was busy caring for her siblings.

As we try to make sense of Alice's nostalgia, her longing to return captures us powerfully, a longing for the space of girlhood often painted as oppressive. In reminiscing about this lost girlhood, Alice's experience supports the view that 'belonging is about where you long to belong.' Belonging as a girl, for her and others, was not just a stage, but was determined by these daily rituals that gave the processes of growing up meaning (Rowe 2005, p. 28). She, therefore, longs to belong to the space from which she was excluded from in the past.

For Alice's aunt, Syokau, now in her late 90s and widowed for over 65 years, participating in our research was a triumph over witchcraft, which affected her childhood. She still remembers how she was taken back to her parents due to infertility and how the elder who was tasked with taking her back went about announcing all over the village that she was returning a barren woman. She laughs derisively at the rituals the man performed after he returned her home and the number of rituals she had to undergo, calling it the corruption of tradition. However, she does not see tradition as always oppressive. She finds solace in the way her father accepted her. The children she bore later in life remind her of the need to be strong. Here again, we note how she overrides the pain of growing up with the joy of the present.

Similarly, 88-year-old Atieno has endured great psychological torment because of her infertility. She left her first marriage because her husband's family tormented her on account of her inability to conceive, and she remarried a man with a wife and children. However, when her husband died, life became difficult for her, and she was taken in by her younger sister Minji, who takes total care of her. Atieno's narrative illustrates the significant burden, stigma, and shame of childlessness, powerfully captured in her remark that her infertility negates her womanhood and confers a disability that is highly stigmatised. Still, the joys of having a supportive sister, brother-in-law, nieces and nephews, a place of her own, and the total embrace of the community she lives in continue to give meaning to such lives that seemed abstracted by tradition.

Wanzoo, who was married to an absentee husband, is happy. While her future one day promised by her stepmother seems to have arrived, this future is still a continuing present, since she is still waiting for her husband to return from Tanzania. She is, however, happy that her children do not fight with their wives. Lack of agency should not be judged from the way she did not resist a forced marriage then, but from the way she positions resistance and resilience as a lifetime event:

> 'When I eventually got my children, I brought them up according to my upbringing. During our days, children belonged to the community/society. My children have not given me a hard time, and remember I was alone.'

Drawing from Wanzoo's narrative above, and in line with our storytelling approach of reading the present through the past, we also go back in time to re-read her agency. We see that resistance for women around arranged marriages, like in her case, is not necessarily a form but an emergence, or what Hughes (2019, p. 1143) sees as 'tracing resistance in its

becoming.' Like Hughes's argument, seeing resistance as emergent, unmooring it from intentionality, and using time to read it helps us avoid foreclosure in our understanding of what it means to resist or to endure in specific spaces. In Wanzoo's case, such resistance is not always predetermined, and spans diverse temporalities where future claims can also be made of particular actions or inactions in the past, like hers (see Hughes 2019, pp. 1142–52).

We move to another participant, Calsina. Married through abduction and denied formal education, her narrative seems to imply that colonial education was not all that was needed for a promising future. Like several older women in our research, she emphasises that she is happy since she is a vital member of the women's self-help group she belongs to, which is successfully providing care for vulnerable children. These women position this group, and the sisterhood they share, as the 'husband' to widows. During a field revisit by the first author, the group meeting was held in her house because she had recently passed on. One of her children was recruited to the group to take her place. A few months after the fieldwork for this study, we were told a church lay leader, Japuonj, Calsina's 60 year old neighbour, with whom we would corroborate perspectives of the church over the years, was 'also no longer with us'.

Minji is in her early 80s. She is our peer researcher on whom and for whom this project was conducted. She depicts another variation of marriage, a love match to her childhood sweetheart, complete with traditional and church wedding ceremonies followed by a long, happy marriage. Denied an education in her adolescence, she positions herself as a survivor of daughter discrimination who was withdrawn from school because the prevailing belief was that 'the only education a girl needed is fetching water, firewood and cooking'. In our discussions, she gave meaning to her life by clapping back on some of the unequal practices she experienced while growing up and as a mother of five daughters. She narrates how she strived to provide the educational benefits that she was denied, first to her children, then relatives, and others in the community. She takes pride in successfully mentoring over 1,200 vulnerable boys and girls within the community. During the fieldwork for this study, she was paying the school fees for three girls she was fostering. She was also the patron of the women's self-help group and mentored young people in the community. This was in addition to supporting her widowed, childless elder sister. In our day-to-day interactions, she was at home (Antonsich 2010), as a woman and as a member of her community, due to her transformative legacy as a fierce advocate and benefactor of education and youth wellbeing.

Minji has also been the rock of our response data for years, and through her, we have sounded our ideas to clear up our uncertainties about indigenous knowledge. Over the years, when unsure, she would always undertake to ask her peers on our behalf. Minji suddenly passed away in 2023, numbing us to the core. Numbing our fingers, the pain of refusal to write the research as yet, at pain with everyone and the earth. Aware that the biographical aspects also impact the time difference between when research is carried out on a topic and when it is published (Puwar 2021), we stopped to care for ourselves and each other. We paused instead to write about the affective relations, which make researchers part of the rhythm, with tears and joys, in the lands they research. Relations that outlast an ethnographic encounter and distort the linear temporalities of 'the before', 'the during', and 'the after' of ethnographic distinctions.

It was in Minji's compound, under a big indigenous tree where, for years since 2008, during our research missions, we would be woken up by the societies of African birds, fighting, co-existing, making beautiful melodies and other stuff only known to the birds. Under the tree, we also listened to the voices in the wind, becoming part of the soul of Siaya County, our research site (Ngutuku 2022). While the birds provided a literal wake-up call each morning, we assert that Minji's departure was a wake-up call to the academy. Calling us to the decolonisation of the academy and its practices, which do not have the language for these affective relations, that make one part of the rhythm with joy, but also with tears and the land.

Despite these key women leaving us, we are emboldened by the realisation that their perspectives were heard and are preserved for posterity, given voice through this research. The reminder by the late Malian philosopher, Amadou Hampâté Bâ, remains more relevant to us than ever when he asserts, 'Each time an old person dies in Africa, a library burns down'.

## 10. In Conclusion

This research aimed to provide a space for women to narrate their stories about coming of age in colonial Kenya. As demonstrated, these stories are often ignored in the knowledge production processes. These narratives have also been rewritten and refashioned by those with the power to know and through their filters. Our approach to researching through storytelling as an African decolonial practice and ontology enabled these women not only to remember their past, but also provided them with a forum for re-imagining their identity, being, and belonging. We have shown that as the women narrated themselves, they also enacted their personhood and individuality as adolescent girls of the past, and as women in the continuing present. Their narratives as an intervention can also be read as their hope to be re-membered otherwise.

The accounts we have presented engage dominant accounts of coming of age, where the voices of women are either silenced or only feature through their relationship to colonialism. We have revealed how an assemblage of other factors, including gender, tradition, girls' agency, and other life characteristics, like poverty and family situation, influence the lived experience of women.

In going beyond the victim narratives, these women exercised an agency that can be read as emergent. They sometimes engaged tradition on its own terms and used human and non-human others, including hope we have read from the future. These forms of being with their world in the historical period, and as read in their continuing present, are not always aligned with Western liberal notions of the personhood of atomic individuals acting on their own. Women's forms of being and becoming with their world and their agency show the contradictions and messy realities of their experiences.

We have also revealed that culture and tradition are sites of contestation, where diverse interpretations were possible in women's accounts of coming of age. Women's accounts, their lived experiences, agency, and performance of being and becoming adolescents might sometimes be read as an impossibility in representing the adolescent girl of the past. While we may not reconcile these contradictions, we can listen to the narrators. We emphasise that the act of listening to the stories of these women is an end in itself. It can unsettle the settled thinking about these vital spaces of growing up in the global South in the past. Such unsettling was a key aim of this research.

**Author Contributions:** Conceptualization, E.N. and A.O.; Methodology, E.N. and A.O.; Formal analysis, E.N. and A.O.; Investigation, E.N. and A.O.; Resources, E.N. and A.O.; Data curation, E.N. and A.O.; Writing—original draft, E.N.; Writing—review & editing, E.N. and A.O.; Visualization, Elizabeth Ngutuku; Supervision, E.N.; Project administration, E.N. and A.O.; Funding acquisition, E.N. and A.O. All authors have read and agreed to the published version of the manuscript.

**Funding:** This collaborative research was supported by ESRC/GCRF-funded Centre for Public Authority and International Development (CPAID) at the London School of Economics and Political Science, and The Research Innovation Facility (RIF) of the International Institute of Social Studies of Erasmus University Rotterdam in the Netherlands.

**Institutional Review Board Statement:** The study was conducted in accordance with the Declaration of Helsinki, and approved by the London School of Economics and Political Science research and ethics committee on 02/11/2021, approval number 47860.

**Informed Consent Statement:** Informed consent was obtained from all participants involved in the study.

**Data Availability Statement:** The datasets presented in this article are not readily available because the research on the lived experience of these women was highly contextual, some based on participant

observation and likely not constructive for sharing or usage by researchers who were not present during its collection. Some general raw data supporting the conclusions of this article will be made available by the authors on request.

**Conflicts of Interest:** The authors declare no conflict of interest.

## Notes

[1]   With each successive birth of yet another daughter, she (and even her mother) endured from her in-laws some vicious identity-based taunts and accusations of filling the home with 'prostitutes' and 'wildcats', and thus 'killing the homestead', which would be reclaimed by the forest in the absence of boys, because girls do not belong, and would get married and leave.

[2]   When the owner of a *simba*—bachelor hut—married, they moved to the huts of the other unmarried sons of the home.

[3]   Musandu (2012) talks about female husbands and, in some contexts, the women married this way among the Akamba were referenced as the wife of the women and, in some cases, as the daughter-in-law. Among the Luo, an infertile woman *Mgumba* could bring a younger woman, typically a younger sister or, preferably, a paternal cousin, for her husband to marry and bear children with. Male infertility, *Buoch*, was legitimate grounds for divorce but could also be masked by advising/persuading the wife to have sexual relations with one of his brothers or cousins to beget children. Children from such a union legitimately belonged to her husband.

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
