# Peer review of "Beyond Colonial Politics of Identity: Being and Becoming Female Youth in Colonial Kenya"

_genealogy, doi:10.3390/genealogy8020047_

Round 1

Reviewer 1 Report

Comments and Suggestions for Authors

This paper exploring the experiences of adolescent girls/young women in colonial Kenya through the lens of Africanist methodologies of dance and performance is compelling reading.  The authors' positionality and situated experiences embodies and embroiders the narratives of the mature Kenyan women re-visiting their experiences of contested childhood, gender/ woman hood, identity, spaces and culture.  

Two points to consider 

1. While the ethics of ownership, validating and giving voice to the narratives of African Women is considered throughout the paper, was there any formal ethical process applied at the inception of the research?

2. Perhaps presentation of the findings/ narratives are a little too long and could be made more concise  in section 7. Gendered notions of sexuality and 8. Marriage practices.    However, I will leave this for the authors to consider.

In sum, this paper was a pleasure to read and makes an invaluable contribution to the giving prominence to the voices and experiences of Kenyan women in (post) colonial Kenya.

Author Response

See Specific Attachment

Reviewer 2 Report

Comments and Suggestions for Authors

This study explores how colonialism impacted the adolescence of Akamba and Luo women in Kenya. Using in-depth interviews, it examines how traditions and colonial forces shaped their identities. The unique "embodied storytelling" approach empowers these women to share their experiences, challenging the victimhood narrative often applied. The authors explain how factors like gender, age, poverty, and family intertwined influence adolescent lives. Instead of portraying them as passive victims, the authors highlight their subjects’ agency, even if it was sometimes limited or contradictory. Ultimately, it calls for a more nuanced understanding of girlhood and coming-of-age experiences in specific historical contexts, particularly in the global South.

The manuscript would benefit from a clear depiction of the research design, setting and data collection. I would also be happy if the authors clarified the significance of their intellectual contribution to the literature on coming-of-age in the contemporary global South. I also found numerous inconsistencies, errors and other issues that require the authors’ attention and clarification. They are listed below.

Line 32: The comma before "identity" is unnecessary and disrupts the flow of the sentence. It should be removed.

Line 40: The quotation from Mignolo is okay, but what is meant by "praxis of living". It could be clarified for better comprehension.

Line 57 and also line 259: Consider clarifying the meaning of "interstices" for readers who may not be familiar with the term.

Lines 113-114: The phrase "in these spaces" lacks clarity and context. It should be revised to specify which spaces are being referred to.

Lines 120-121: There is a repetition of "the" before "historical legacy."

Line 150: The phrase "misgendered girls" may need clarification depending on the context and intended meaning.

Line 168: Consider clarifying or expanding upon the significance of Bellows-Blakely's perspective to provide context for the reader.

Line 184: Consider clarifying what is meant by "re-gendering the epistemic enterprise of knowledge production" for readers who may not be familiar with this terminology.

Lines 195-197: "And while still haunted by these descriptions..." - Consider revising the sentence for better clarity and coherence.

Lines 205-209: “By drawing on…” This sentence is complex and could benefit from being broken down or simplified for easier understanding.

Line 229: Consider inserting a comma after “as we reveal”.

Lines 232-235: “For a category like female youth…” This sentence is quite long and could be broken down for better readability.

Line 236: Consider clarifying the role of Deleuze's concept of becoming in the context of resilience.

Line 265: Delete an extra space before “in Jera”.

Line 275-276: This sentence could be clarified by explaining what is meant by "re-storying" and "reframing narratives" for readers who may not be familiar with these terms.

Lines 298-299: "We also did repeat life history interviews…" - Consider providing more information about the selection criteria for these women and the purpose of conducting repeat interviews.

Lines 300-301: “For perspectives on generational differences…" - Consider elaborating on why perspectives from older individuals were sought and how they contribute to the overall understanding of the research topic.

Lines 311-312: Consider clarifying what is meant by “triangulation”.

Lines 334-335: "After obtaining oral consent..." - Consider mentioning whether the translations were done by professional translators or by the researchers themselves, as this could affect the accuracy of the translations.

Lines 349-351: “We were also particularly aware..." - This sentence could be clarified by explaining what is meant by "cognitive empire" and how it relates to the researchers' positionality.

Lines 367-369: "For example, in the Luo research site..." - Consider rephrasing for clarity, perhaps separating this sentence into two to avoid confusion.

Lines 370-373: "Similarly, in Akamba, Alice…'" - Consider providing more context about the significance of Alice's question and its implications for the research process.

Lines 374-375: "In taking our cues from St. Pierre's view that..." - Consider clarifying who St. Pierre is and providing a brief explanation of her concept of "response data" for readers who may not be familiar with it.

Lines 386-387: Consider elaborating on the perils of cultural insiderism and providing examples or evidence to support this assertion.

Lines 415-416: "Focus group discussions revealed..." - Expand on how maturity was defined and recognized within the community beyond marriage, and how this perspective differed from conventional notions of maturity.

Lines 457-458: Elaborate on the significance of guidance and conversations around sexual maturation, including menstruation management, and how these practices were connected to broader cultural beliefs and environmental conservation efforts.

Lines 462-465: Provide more detail on the symbolic actions and rituals performed during initiation ceremonies, such as singing and dancing in a calculated posture among the Kamba, and how these actions reflect cultural values and relationalities.

Lines 485-487: Provide more detail on the communal aspects of guidance and learning, such as the communal harvesting of food among the Kamba and the group dating practices among the Luo. Exploring the social dynamics and cultural norms surrounding these activities would enrich the reader's understanding.

Lines 513-516: Consider explaining the societal norms and values that underpin these proverbs and the broader cultural beliefs about masculinity and femininity.

Lines 566-567: “…is seen as notions of relative chastity”. Seen by whom? Insert a space after “chastity”.

Lines 576-578: Consider clarifying the meaning of " panoptical gaze" for readers who may not be familiar with the term.

Line 608: What is meant by “situated agency” and how is this term different from just “agency”?

Lines 624-626: I would like to hear more on the implications of Alice's capitulation to the dominant norms for her sense of identity and self-worth.

Line 651: Delete an extra space before “Kenya”.

Line 652: Provide more detail on “colonial modes of accumulation”.

Line 667: Delete an extra space after “and”.

Line 686: Delete an extra space after “where”.

Line 692: Delete an extra space after “least”.

Line 695: Delete an extra space before “Esther”.

Lines 704-705: “According to Jacob (1961)…” is a stand-alone sentence. Consider merging it with the paragraph that follows.

Line 714: Delete an extra space before “7”.

Line 735: Consider clarifying the meaning of "life forces and intensities" for readers who may not be familiar with the terms.

Line 736: Delete an extra space before “Loew”. Insert a full stop after “2012)”.

Lines 795-797: Consider the implications of the vetting process for power relations within marriage.

Line 818: Delete an extra space after “Iweto”.

Line 850: "Alice had, for example narrated…". Consider instead: "For example, Alice had narrated".

Line 901: Delete an extra space after “Calsina.”

Lines 919-920: "Together with her husband, they were trailblazers educating their children..." – is grammatically incorrect. It should be "Together with her husband, she was a trailblazer educating their children..."

Line 929: "Aware that the biographical..." The phrase "impact on" should be "impact the" for correct grammar.

Lines 935-936: "Distorted temporalities…" This sentence is unclear and could be revised for clarity.

Line 964: Delete an extra space before “tradition”.

Lines 977-978: "Listening can unsettle…" - The phrase "settled thinking" could be clarified or rephrased for better clarity. Also, insert a full stop at the end of the sentence. 

Reviewer 3 Report

Comments and Suggestions for Authors

I can see why the other two readers were split, as this essay has considerable strengths but also many weaknesses. its strengths are: it is empirically in touch with its situation; it has a combination of 'insider and 'outsider' knowledge, and it is vigorously and transparently argued. its weaknesses are that it is severely under-theorized, and statements are made that are not ballasted by abstract conceptualizations. I have made detailed comments on the PDF that I hope will aid the authors as they revise. Although I am asking the authors to do a good bit of work, I am choosing the 'slight revision' option because I believe that all the authors have to do is conceptualize more, even if they have to do this at several important places. 

Comments on the Quality of English Language

Generally good, only minor issues which I ave pointed out in the PDF. 

Round 2

Reviewer 2 Report

Comments and Suggestions for Authors

It seems like the authors have made some improvements to the paper revisions! They have improved the clarity and quality of the content.

Author Response

We appreciate the comment from the reviewer